# The Origin of Late Roman Period–Post-Migration Period Lithuanian Horses

**Giedrė Piličiauskienė** [1,*], **Laurynas Kurila** [1], **Edvardas Simčenka** [1], **Kerstin Lidén** [2], **Ellen Kooijman** [3], **Melanie Kielman-Schmitt** [3] and **Gytis Piličiauskas** [4]

1  Department of Archaeology, Vilnius University, Universiteto st. 7, 01513 Vilnius, Lithuania; laurynas.kurila@istorija.lt (L.K.); edvardas.simcenka@istorija.lt (E.S.)
2  Archaeological Research Laboratory, Department of Archaeology and Classical Studies, Stockholm University, SE-10691 Stockholm, Sweden; kerstin.liden@arklab.su.se
3  Department of Geosciences, Swedish Museum of Natural History, Box 50 007, SE-104 05 Stockholm, Sweden; ellen.kooijman@nrm.se (E.K.); melanie.schmitt@nrm.se (M.K.-S.)
4  Archaeology Department, Lithuanian Institute of History, Tilto st. 17, 01101 Vilnius, Lithuania; gytis.piliciauskas@gmail.com
*  Correspondence: giedrepils@gmail.com; Tel.: +37-061-018-389

**Abstract:** In this paper, we present the $^{87}Sr/^{86}Sr$ data of 13 samples from horses from six Lithuanian burial sites dating from the 3rd to the 7th C AD. Alongside these data, we also publish the bioavailable $^{87}Sr/^{86}Sr$ data of 15 Lithuanian archaeological sites, based on 41 animals which enabled the construction of a reliable baseline for the Southeast Baltic area. The $^{87}Sr/^{86}Sr$ values partially confirmed the hypothesis that the unusually large horses found in Late Roman Period to Post-Migration Period burials are of non-local origin. Of the three non-local horses identified, two were among the largest specimens. However, the overlap of bioavailable $^{87}Sr/^{86}Sr$ data across different European regions does not permit us to establish whether the non-local horses originated from other areas in Lithuania or from more distant regions. With regards to the $^{87}Sr/^{86}Sr$ data, the place of origin of the non-local horses could be Southern Sweden. This encourages discussions on the possible directions of migration and compels us to rethink the current models that posit South and Central Europe as the main sources of migration. The results of the $^{87}Sr/^{86}Sr$, $\delta^{13}C$, and $\delta^{15}N$ analyses demonstrate that horses buried in the same cemetery had different mobility and feeding patterns. Differences could be due to the different function and sex of the horses as well as the lifestyle of their owners. The most sedentary horses were pregnant mares, while the extremely high $\delta^{15}N$ of three horses may reflect additional fodder and probably a better diet.

**Keywords:** $^{87}Sr/^{86}Sr$ analysis; horses; migration; mobility; Roman and Migration Period; Southeast Baltic

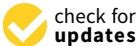



## 1. Introduction

The tradition of burying horses—the whole animal or its parts—in graves together with humans had been a widespread practice throughout Eurasia for several millennia [1–3]. This practice is closely connected to warfare, military elites, and is especially frequent during periods of migration and conflict. The rite of horse burials spread across Lithuania in the first centuries AD and persisted until the Christianisation of Lithuania in the late 14th C AD [4–6]. The human and horse burials during the mid to 1st millennium stand out above all of the other human and horse burials in terms of wealth, imported grave goods, and the large number of weapons. Their appearance during the 3rd to 7th C AD (the Late Roman Period (LRP), Migration Period (MP), and Post-Migration Period (PMP)) gives rise to discussions regarding a possible non-local origin of the horses. Furthermore, both the general archaeological record and the horse graves themselves indicate at least two different cultural influences. The appearance of warrior burials with horses in barrow

mounds in Eastern Lithuania is associated with the Middle Danube region [5]. To date, only nine of these burials have been excavated. In the Western and Central Lithuanian cemeteries, the buried human remains were accompanied by whole horses or only their heads and legs [6]. This latter tradition spread into Lithuania from the West via the Sambia Peninsula (modern Kaliningrad, Russian Federation). Currently, about 150 of these graves are known in Lithuania. A similar tradition or partial horse burial or sacrifice also reached Scandinavia, while its origin is attributed to Asian nomads [7].

With regards to the data of previous zooarchaeological analyses [8], a large part of LRP–PMP horses in Lithuania were taller, 15–25 cm, than the horses during the Viking Period (VP, 9th–11th C AD) and the Medieval Period (12th–14th C AD). Moreover, LRP–PMP horses are frequently found in graves of exceptionally rich males who were buried with grave goods of non-local origin. This evidence led to the hypothesis that the "large" horses might also be of non-local origin and that they were brought to Lithuania by warrior groups which participated in military conflicts that occurred during the MP. In general, horses are highly mobile animals, that in the past were transported on the hoof for large distances as means of transport, tradeable goods, spoils of war or as expensive gifts [9,10]. After a re-evaluation of the osteometric data of LRP–PMP horses showing that the size differences between LRP–PMP and later periods' horses are slightly smaller, it became clear that the data of earlier analyses were also in need of re-examination. However, a significant part of the 4th–7th C AD horses was still unusually large for Lithuania: The average height of LRP–PMP horses was 130.2 cm, VP horses was 123.6 cm, and 12th–14th C AD horses was 127.8 cm. A few of the LRP–PMP individuals are osteometrically similar to the horses found in the Rhine–Danube region, South Scandinavia, and the Balkans [11]. The size of horses as well as imported finds from the graves allow us to presume that there may be links between the LRP–PMP horses found in Lithuania and the large horses that spread through Europe, with the expansion of the Roman Empire and after the fall of the Western Roman Empire [12,13].

When considering the exceptionally large LRP–PMP horses buried alongside grave goods, we noted that some of these burials were analogous to those found in the lands of different Germanic tribes. Through these observations, we formulated three principal hypotheses regarding the origins of people and horses from this period in Lithuania: (1) The individuals were wealthy non-local warriors who reached the territory of the modern-day Lithuania together with their horses from Southern, Western or Northern Europe; (2) the individuals were local people, who joined the armies on distant campaigns and who afterwards returned to their native land with their non-local horses; and (3) the individuals were local people who acquired non-local goods and horses. To date, all of these hypotheses could be discussed only based on archaeological data and could not be proven or dismissed solely on this basis.

To test our hypotheses, we conducted the $^{87}Sr/^{86}Sr$ analysis on samples from horse specimens from the burials as well as created a $^{87}Sr/^{86}Sr$ baseline based on animal specimens from local archaeological sites. The $^{87}Sr/^{86}Sr$ analysis has already been successfully applied in studies on past human and animal mobility for over three decades, e.g., [14–17]. Different rocks exhibit different $^{87}Sr/^{86}Sr$ ratios that do not alter when the strontium is transferred from the weathered rocks to surface water, soil, and vegetation. Biologically available $^{87}Sr/^{86}Sr$ from food and water is incorporated into animal tissues, including tooth enamel, which forms at a young age and does not reconstitute after mineralization. The $^{87}Sr/^{86}Sr$ analysis has already been implemented to identify the origin of MP horses sacrificed in the Illerup Åadal bog in Denmark [13]. In the East Baltic region, the values of biologically available $^{87}Sr/^{86}Sr$ are to date only known from Estonia [18,19], Latvia based on measurements on three snail shells, and for Northwestern Russia [19]. To date, there are no $^{87}Sr/^{86}Sr$ baseline data from Belarus or Northeast Poland. Baseline data from these regions would be of great importance for contextualizing the data from Lithuania.

In this paper, we publish a baseline for the Southeast Baltic for the bioavailable $^{87}Sr/^{86}Sr$ data based on 41 animals from 14 archaeological sites in Lithuania. Moreover, we

publish the $^{87}$Sr/$^{86}$Sr ratios of 13 horses dating to the 3rd–7th C AD from six Lithuanian burial sites. To date, only 21 horse skeletons of sufficient preservation, dating to the LRP–PMP, have been recovered in Lithuania. Therefore, the 13 individuals analyzed comprise 61.9% of all the horses from the period of interest in this study, a period associated with a military elite. The results of the analyses provided insights into the origin and mobility, as well as on the herding practices, of the horses from the period.

## 2. Material and Methods

### 2.1. $^{87}$Sr/$^{86}$Sr Analysis

We performed the $^{87}$Sr/$^{86}$Sr analysis on 41 animal tooth enamel samples from 14 archaeological sites (Figure 1), to provide a baseline for bioavailable $^{87}$Sr/$^{86}$Sr in Lithuania. However, in order to increase the precision and reliability, we obtained permission to incorporate the $^{87}$Sr/$^{86}$Sr data of archaeological animals analyzed as part of another project currently carried out in Lithuania [20]. We selected the samples of the archaeological animals for the baseline from settlement sites that were as close as possible to the burial sites of horses, both in terms of distance and chronology. The tooth enamel of animals with small home ranges was preferably analyzed, where possible (Table 1 and Table S1: 18 and [20]).

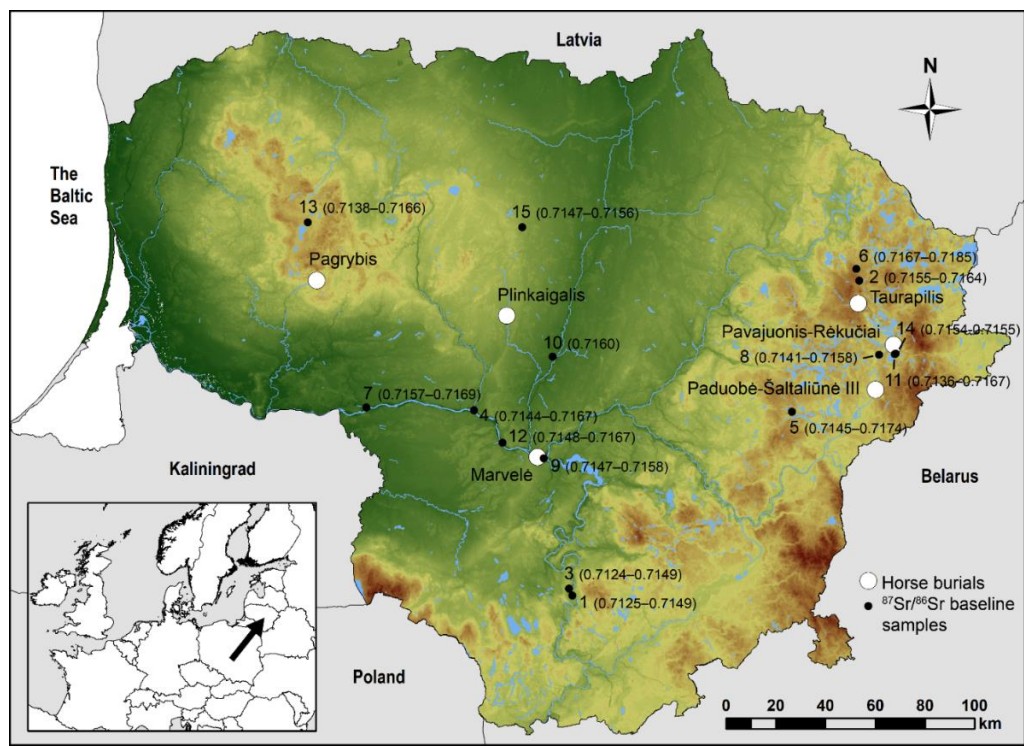

**Figure 1.** Locations of horses and animal samples analyzed for this study and incorporated from another study [20]: 1. Alytus; 2. Antilgė; 3. Bakšiai; 4. Daučionys; 5. Dubingiai; 6. Garniai; 7. Jurbarkas; 8. Kaltanėnai; 9. Kaunas; 10. Kėdainiai; 11. Kretuonas 1C, 1D; 12. Kulautuva; 13. Varniai; 14. Žemaitiškė 2; 15. Žilionys (© National Land Service under the Ministry of Agriculture of the Republic of Lithuania).

In total, 15 tooth enamel samples from 13 horses have been analyzed. For 11 individuals, one permanent tooth was analyzed, while for the remaining two individuals, two teeth—a permanent and a deciduous—were analyzed (Table 2 and Table S1: 3–17). Five to 38 samples have been analyzed from a single tooth, starting from the highest point of the crown and then proceeding downwards towards the root. Enamel mineralization begins from the upper part of the tooth [21]. Therefore, the first samples from the upper part of

the tooth reflect a younger age of the horse than the samples taken from the lower part of the tooth. Depending on the tooth, mineralization occurs over a period of ~2 weeks to ~4.5 years and reflects the individual's environment and diet during the period of mineralization in the tooth analyzed [21]. From the permanent molars, $P_2$ (mineralization period of ~13–31 months) and $M_2$ (mineralization period of ~7–37 months) teeth were used in the study [21]. We were unable to find comparable data from other studies on the mineralization time for permanent incisors in horses. However, knowing their eruption time and the mineralization patterns of the molars [21–23], it is possible to establish that the analyzed permanent incisors ($I_1$, $I_2$, $I_3$) reflect the environment of a young horse less than 3.5–5 years old. Mineralization of deciduous teeth, depending on the tooth, begins during the 5th to 8th month of pregnancy [23]. Their enamel is mineralized before birth [22]. Therefore, the $^{87}Sr/^{86}Sr$ ratio of deciduous teeth gives an indication of the pregnant mare's environment and mobility. Three different deciduous teeth were used for the study: $dI_1$, $dI_3$, $dP_4$.

The $^{87}Sr/^{86}Sr$ ratios from the tooth enamel samples were measured at the Vegacenter facility of the Swedish Museum of Natural History (Stockholm, Sweden) using an ESI NWR193 ArF excimer based laser ablation system (Electro Scientific Industries, Portland, OR, USA) coupled to a Nu Plasma (II) multi-collector inductively coupled plasma mass spectrometer (Nu Instruments Ltd., Wrexham, UK). The instrument operating conditions are listed in Table S1: 1 and 2, and the stable isotope data for each ablated tooth are provided in Table S1: 3–18. The method follows the procedures outlined in previous studies [16,24].

All of the analyzed horses were radiocarbon dated [11]. The results revealed that some burials previously suggested to date to the Migration Period, are actually earlier or later, dating to the Late Roman or Post-Migration Period.

*2.2. $\delta^{13}C$ and $\delta^{15}N$ Analysis*

The $\delta^{13}C$ and $\delta^{15}N$ stable isotopes analysis was undertaken at the Centre for Physical Sciences and Technology, Vilnius (Lithuania). During sampling, we preferred mandible or cranium, from which the tooth for the $^{87}Sr/^{86}Sr$ analysis was also taken. In some cases (Plinkaigalis 3A, 3B, Pagrybis 145, 157), metacarpal and metatarsal bones were sampled, when the first samples did not meet the collagen quality criteria. Bone collagen extraction was performed according to the acid–alkali–acid (AAA) procedure followed by gelatinization [25]. Samples were treated with 0.5 M hydrochloric acid and 0.1 M sodium hydroxide. Bone collagen gelatinization was performed in a pH 3 solution at 70 °C for 20 h. Gelatin solution was filtered using a cleaned Ezee-filter and freeze-dried. For $^{14}C$, $\delta^{13}C$, and $\delta^{15}N$ measurements, the same collagen aliquot was used. Stable carbon and nitrogen isotope ratio values in the bone collagen were measured using an elemental analyzer (Thermo FlashEA 1112) connected to an Isotope Ratio Mass Spectrometer (Thermo Finnigan Delta Plus Advantage). The analytical precision for $\delta^{13}C$ and $\delta^{15}N$ was ± 0.1‰ and ± 0.15‰, respectively. Stable isotope data are reported as δ values in permille (‰) relative to international standards: Vienna Pee Dee Belemnite (V-PDB) for $\delta^{13}C$ and Ambient Inhalable Reservoir (AIR) for $\delta^{15}N$. To ensure that the collagen is of sufficient quality, four collagen quality parameters were also measured in the bone samples and used as indicators of collagen integrity [26–28]. The animal remains that were analyzed, are stored in the Zooarchaeological Repository of Vilnius University, Faculty of History, except for the horse from Paduobė-Šaltaliūnė III, which is stored in the National Museum of Lithuania.

**Table 1.** $^{87}$Sr/$^{86}$Sr measurements for animal teeth enamel (subneolithic—5000–2900 cal BC; Early Bronze Age—1800–1100 cal BC; Late Bronze Age—1100–600 cal BC; Early Modern Period—16th–19th C AD).

| No. | Burial Site | Distance from Baseline Site | Baseline Site | Period | Sample No. | Animal | 87Sr/86Sr Mean | SD | 2SD |
|---|---|---|---|---|---|---|---|---|---|
| 1 | | | Kaunas | Medieval | 33 | roe deer (*Capreolus capreolus*) | 0.7158 | 0.0003 | 0.0005 |
| 2 | | 1.5 km | Kaunas | Medieval | 34 | sheep/goat (*Ovis aries/Capra hircus*) | 0.7153 | 0.0002 | 0.0004 |
| 3 | Marvelė | | Kaunas | Medieval | 35 | pig (*Sus scrofa domestucus*) | 0.7147 | 0.0002 | 0.0004 |
| 4 | | | Kulautuva | Migration—Medieval | 30 | roe deer (*Capreolus capreolus*) | 0.7159 | 0.0010 | 0.0020 |
| 5 | | 15 km | Kulautuva | Migration—Medieval | 31 | pig (*Sus scrofa domesticus*) | 0.7167 | 0.0002 | 0.0005 |
| 6 | | | Kulautuva | Migration—Medieval | 32 | sheep/goat (*Ovis aries/Capra hircus*) | 0.7148 | 0.0001 | 0.0002 |
| 7 | | 7 km/13 km | Kaltanėnai | Subneolithic—Roman | 9 | beaver (*Castor fiber*) | 0.7158 | 0.0001 | 0.0001 |
| 8 | | | Kaltanėnai | Subneolithic—Roman | 10 | beaver (*Castor fiber*) | 0.7141 | 0.0006 | 0.0011 |
| 9 | | | Žemaitiškė 2 | Subneolithic—Early Bronze | 7 | red deer (*Cervus elaphus*) | 0.7154 | 0.0002 | 0.0004 |
| 10 | Paduobė-Šaltaliūnė III Pavajuonis-Rėkučiai | | Žemaitiškė 2 | Subneolithic—Early Bronze | 8 | beaver (*Castor fiber*) | 0.7155 | 0.0006 | 0.0012 |
| 11 | | | Kretuonas 1C | Subneolithic | 1 | marten (*Martes martes*) | 0.7140 | 0.0002 | 0.0003 |
| 12 | | 15 km/3.5 km | Kretuonas 1C | Subneolithic | 2 | marten (*Martes martes*) | 0.7147 | 0.0005 | 0.0009 |
| 13 | | | Kretuonas 1C | Subneolithic | 3 | beaver (*Castor fiber*) | 0.7201 | 0.0010 | 0.0020 |
| 14 | | | Kretuonas 1C | Subneolithic | 4 | beaver (*Castor fiber*) | 0.7151 | 0.0007 | 0.0014 |
| 15 | | | Kretuonas 1C | Subneolithic | 5 | roe deer (*Capreolus capreolus*) | 0.7155 | 0.0001 | 0.0003 |
| 16 | | | Kretuonas 1C | Subneolithic | 6 | roe deer (*Capreolus capreolus*) | 0.7155 | 0.0003 | 0.0006 |
| 17 | Pagrybis | 25 km | Varniai | Early Modern | 40 | cattle (*Bos taurus*) | 0.7155 | 0.0002 | 0.0003 |
| 18 | | | Varniai | Early Modern | 45 | cattle (*Bos taurus*) | 0.7138 | 0.0006 | 0.0011 |

**Table 1.** *Cont.*

| No. | Burial Site | Distance from Baseline Site | Baseline Site | Period | Sample No. | Animal | 87Sr/86Sr Mean | SD | 2SD |
|---|---|---|---|---|---|---|---|---|---|
| 19 | | | Žilionys | Roman—Medieval | 28 | pig (*Sus scrofa domesticus*) | 0.7156 | 0.0001 | 0.0003 |
| 20 | | | Žilionys | Roman—Medieval | 29 | pig (*Sus scrofa domesticus*) | 0.7151 | 0.0001 | 0.0001 |
| 21 | Plinkaigalis | 40 km | Žilionys | Roman—Medieval | 27 | sheep/goat (*Ovis aries/Capra hircus*) | 0.7147 | 0.0001 | 0.0003 |
| 22 | | | Daučionys | Migration—Medieval | 24 | roe deer (*Capreolus capreolus*) | 0.7164 | 0.0006 | 0.0012 |
| 23 | | | Daučionys | Migration—Medieval | 25 | sheep/goat (*Ovis aries/Capra hircus*) | 0.7163 | 0.0003 | 0.0006 |
| 24 | | | Garniai | Late Bronze | 11 | marten (*Martes martes*) | 0.7180 | 0.0011 | 0.0021 |
| 25 | | 14 km | Garniai | Late Bronze | 12 | pig (*Sus scrofa domesticus*) | 0.7167 | 0.0002 | 0.0003 |
| 26 | Taurapilis | | Garniai | Late Bronze | 13 | fox (*Vulpes vulpes*) | 0.7185 | 0.0006 | 0.0012 |
| 27 | | 10 km | Antilgė | Late Bronze-Roman | 14 | pig (*Sus scrofa domesticus*) | 0.7164 | 0.0006 | 0.0013 |
| 28 | | | Antilgė | Late Bronze-Roman | 15 | pig (*Sus scrofa domesticus*) | 0.7155 | 0.0002 | 0.0004 |
| 29 | | | Alytus | Migration—Medieval | 36 | pig (*Sus scrofa domesticus*) | 0.7133 | 0.0003 | 0.0007 |
| 30 | | | Bakšiai | Roman | 21 | pig (*Sus scrofa domesticus*) | 0.7124 | 0.0004 | 0.0008 |
| 31 | | | Bakšiai | Roman | 22 | pig (*Sus scrofa domesticus*) | 0.7136 | 0.0001 | 0.0001 |
| 32 | | | Bakšiai | Roman | 23 | sheep/goat (*Ovis aries/Capra hircus*) | 0.7149 | 0.0006 | 0.0012 |
| 33 | | | Daučionys | Migration—Medieval | 26 | red deer (*Cervus elaphus*) | 0.7154 | 0.0002 | 0.0005 |
| 34 | | | Dubingiai | Medieval | 16 | pig (*Sus scrofa domesticus*) | 0.7150 | 0.0001 | 0.0003 |
| 35 | | | Dubingiai | Medieval | 17 | pig (*Sus scrofa domesticus*) | 0.7150 | 0.0001 | 0.0003 |
| 36 | Other | | Dubingiai | Medieval | 18 | pig (*Sus scrofa domesticus*) | 0.7145 | 0.0003 | 0.0005 |
| 37 | | | Dubingiai | Medieval | 19 | sheep/goat (*Ovis aries/Capra hircus*) | 0.7174 | 0.0001 | 0.0001 |
| 38 | | | Dubingiai | Medieval | 20 | sheep/goat (*Ovis aries/Capra hircus*) | 0.7152 | 0.0001 | 0.0002 |
| 39 | | | Jurbarkas | Early Modern | 37 | pig (*Sus scrofa domesticus*) | 0.7157 | 0.0005 | 0.0010 |
| 40 | | | Jurbarkas | Early Modern | 38 | sheep/goat (*Ovis aries/Capra hircus*) | 0.7158 | 0.0001 | 0.0002 |
| 41 | | | Jurbarkas | Early Modern | 39 | pig (*Sus scrofa domesticus*) | 0.7169 | 0.0003 | 0.0006 |

*2.3. Site Description*

Marvelė is a burial site in Central Lithuania dating to the 2nd–12th C AD and is the largest excavated burial site in the Baltic States. In 1991–2007, 236 horses and 1591 human graves were discovered, including rich graves of LRP–MP warriors with abundant weapons and other equipment. Horses were buried individually in separate parts of the cemetery. Most of the horses are dated to the VP. However, a small group was assigned to LRP–MP [29,30]. The $^{87}$Sr/$^{86}$Sr analysis of one horse (grave 113) unrelated to any human burial was performed.

Paduobė-Šaltaliūnė III is a barrow cemetery in Eastern Lithuania and is part of a huge array of about 50 burial sites with about 1700 barrows, where graves of both elite warriors and elite females were discovered. In 1987–2008, 18 barrows were excavated. In barrow 17, a burial of a male and of a horse were found, which were plundered. The remaining grave goods are typical for Central, Southern or Western Europe and allow the grave to be relatively dated to the 5th C AD [5,31].

Pagrybis is a cemetery in Western Lithuania, dating to the 4th–7th C AD, that was excavated between 1980–1982. A large number of human graves were found here (217) and of these, 35 males and 3 children were buried with horse parts, including the head and lower limbs. Some of the graves with horses contained numerous warrior grave goods [32,33]. To date, the remains of 10 horses have been survived, and the $^{87}$Sr/$^{86}$Sr ratios of four individuals (graves 104, 145, 157, and 207) have been analyzed. All of these graves were disturbed, likely plundered, but none of the accompanying grave goods indicate the highest social status.

Pavajuonis-Rėkučiai is a barrow cemetery in Eastern Lithuania dated to the 4th–early 5th C AD and is located in the same array of barrow cemeteries as Paduobė-Šaltaliūnė III. It was excavated in 1994 and then again in 1996. One horse was found together with a male in a robbed grave in barrow 6 [5,34]. The possible non-local origin of the human is indirectly indicated by the graves in the adjacent barrow, which contained rich grave goods that were typical for the LRP–MP Gothic culture of the Black Sea coast and Southern and Central Europe [35,36].

Plinkaigalis is a cemetery in Central Lithuania dated to the 4th–7th C AD and was excavated between 1977–1984. A large number of human graves were found here, with a total of 372. The cemetery is extremely rich in grave goods [37]. Some items are of a non-local origin and are associated with the Sambian Peninsula or South Scandinavia [38–40]. Traces of violent trauma in human bones and embedded arrowheads typical of Asian nomads indicate potential conflicts and possible (im)migration. The cemetery contained four horse graves buried separately from humans, of which two horses were buried in grave 3. The $^{87}$Sr/$^{86}$Sr analyses were carried out on three horses (graves 2, 3A, and 3B).

Taurapilis is a barrow cemetery in Eastern Lithuania dated to the 5th C AD that was excavated in 1970–1971. Horses were found in four barrows (1, 4, 5, and 6) and they were buried together with rich and well-armed males. An extremely rich warrior grave in barrow 5 contained a set of weapons, silver and gilded pieces of equipment, and jewellery, typical of various barbarian regions of Southern, Central, and Northern Europe [41]. This is the northernmost grave in Europe of a representative of the military elite of the MP, the so-called "duke". This cemetery is considered as the burial place of a military chieftain and his retinue [5]. The inter-regional origin of the chieftain's grave goods possibly indicates that this group participated in long-distance military campaigns before reaching Lithuania. Three horses found in burials 4, 5, and 6 were analyzed for $^{87}$Sr/$^{86}$Sr.

## 3. Results

The range of bioavailable $^{87}$Sr/$^{86}$Sr data defined for Lithuania, constructed from the archaeological animal remains are 0.7124–0.7187 for inland and 0.7095–0.7148 for coastal areas (Figure 1, Tables 1 and S1, and [20]). The highest $^{87}$Sr/$^{86}$Sr values were identified in Northeastern Lithuania, while the lowest values were identified in Southwestern and coastal Lithuania. For each of the burial sites, individual baselines have been constructed.

For the Marvelė cemetery, we selected animal remains (*n* = 5) from two sites (Table 1). The $^{87}$Sr/$^{86}$Sr baseline estimated for Marvelė is 0.7158 ± 0.00168 (2 SD). The 127.9 cm tall horse buried in Marvelė (grave 113) had $^{87}$Sr/$^{86}$Sr values (0.7141–0.7145), which fit with the strontium baseline for the area surrounding the cemetery (Figure 2: 1; Table 2). Intra-tooth variation SD (0.0001) of this horse is one of the lowest of all of the horses analyzed (Figure 3), implying that this individual had very low mobility at the time of the tooth's formation. Although the strontium analysis suggests that the horse was of local origin, this individual stands out from the horses of local type due to its slightly larger size [6,11].

The barrow cemeteries of Paduobė-Šaltaliūnė III and Pavajuonis-Rėkučiai are separated by a distance of 18 km. However, both are located on the Žeimena Plain, which is covered by alluvial and glaciofluvial sediment. Therefore, a single baseline from the $^{87}$Sr/$^{86}$Sr of faunal remains from three sites was constructed for these barrow cemeteries. The baseline based on 14 animals from this region is 0.7149 ± 0.0016 (2 SD) (Tables 1 and S1: 18 and [20]). In the barrow from Paduobė-Šaltaliūnė III, a 137.7 cm tall horse was buried. The $^{87}$Sr/$^{86}$Sr (0.7186–0.7190) ratios of the first three and the last two measurements made at the crown of the tooth are all higher than the local baseline, whereas the remaining five measurements all fall within it (Figure 2: 5). This individual's intra-tooth variation SD is the highest of all the horses analyzed (0.0011) and reflects a high degree of mobility (Figure 3). This horse is also distinguished by the lowest $\delta^{15}$N value (4.8‰) among the LRP–MP horses in Eastern Lithuania (Table 2; Figure 4). The horse from the Pavajuonis-Rėkučiai barrow cemetery was 129.3 cm tall. Its $^{87}$Sr/$^{86}$Sr value does not deviate from the boundaries of the local baseline (Figure 2: 5). The individual had a low mobility life during the first years of its life, as its intra-tooth variation SD is the lowest among all of the horses analyzed (0.0001). However, the horse stands out due to the unusually depleted $\delta^{13}$C value (−24.0‰) and enriched $\delta^{15}$N (7.4‰) value (Figure 4).

**Table 2.** Horse teeth studied with averaged $^{87}$Sr/$^{86}$Sr measurements. Withers height (WH) was estimated after May [42] and is given in cm. References for chronology [11].

| No. | Burial Site, Grave | Sampled Tooth | Withers Height | Age, Years | AMS 14C cal AD (95.4%) | $\delta^{13}$C | $\delta^{15}$N | C/N | Number of Lines | $^{87}$Sr/$^{86}$Sr Average | SD | Interpretation |
|---|---|---|---|---|---|---|---|---|---|---|---|---|
| 1 | Marvelė 113 | I1 | 127.9 | 9–10 | 134–408 | - | - | - | 10 | 0.7142 | 0.0001 | local |
| 2 | Paduobė-Šaltaliūnė III 17 | P2 | 137.7 | 7–8 | 263-530 | -22,1 | 4.8 | 3.2 | 10 | 0.7172 | 0.0011 | nonlocal |
| 3 | Pagrybys 104 | I1 | 132.3 | 3.5–4 | 580–668 | -22.1 | 4.1 | 3.2 | 30 | 0.7144 | 0.0002 | local |
| 4 | Pagrybys 145 | I1 | 129.3 | 8–9 | 601–662 | -22.6 | 5.5 | 3.3 | 10 | 0.7149 | 0.0002 | local |
| 5 | Pagrybys 157 | I3 | 127.3 | 9–10 | 540–640 | -22.6 | 5.6 | 3.3 | 38 | 0.7162 | 0.0005 | local |
| 6 | Pagrybys 207 | P2 | 125.4 | 7–8 | 565–654 | -22.4 | 7.1 | 3.2 | 5 | 0.7186 | 0.0003 | nonlocal |
| 7 | Pavajuonis-Rėkučiai 6 | P2 | 129.3 | 5–7 | 234–426 | -24.0 | 7.4 | 3.2 | 15 | 0.7160 | 0.0001 | local |
| 8 | Plinkaigalis 2 | dI3 | 127.0 | 4–5 | 585–774 | -23.1 | 4.8 | 3.2 | 16 | 0.7148 | 0.0002 | local |
| | | I3 | | | | | | | 10 | 0.7160 | 0.0005 | local |
| 9 | Plinkaigalis 3A | dP4 | 129.4 | 1.5–2 | 574–651 | -22.5 | 6.0 | 3.3 | 5 | 0.7160 | 0.0002 | local |
| | | M2 | | | | | | | 10 | 0.7158 | 0.0004 | local |
| | | M2 | | | | | | | 8 | 0.7157 | 0.0002 | local |
| 10 | Plinkaigalis 3B | I1 | 137.3 | 7–8 | 600–663 | -22.6 | 7.0 | 3.3 | 23 | 0.7169 | 0.0003 | nonlocal |
| 11 | Taurapilis 4 | I1 | 130.8 | 8–10 | 259–538 | -21.9 | 5.5 | 3.2 | 10 | 0.7168 | 0.0002 | local |
| 12 | Taurapilis 5 | I2 | 125.8 | 3–3.5 | 236–530 | -22.4 | 5.5 | 3.3 | 18 | 0.7152 | 0.0002 | local |
| 13 | Taurapilis 6 | dI2/3 | 127.6 | 3–3.5 | 235–430 | -22.8 | 5.4 | 3.1 | 4 | 0.7153 | 0.0001 | local |

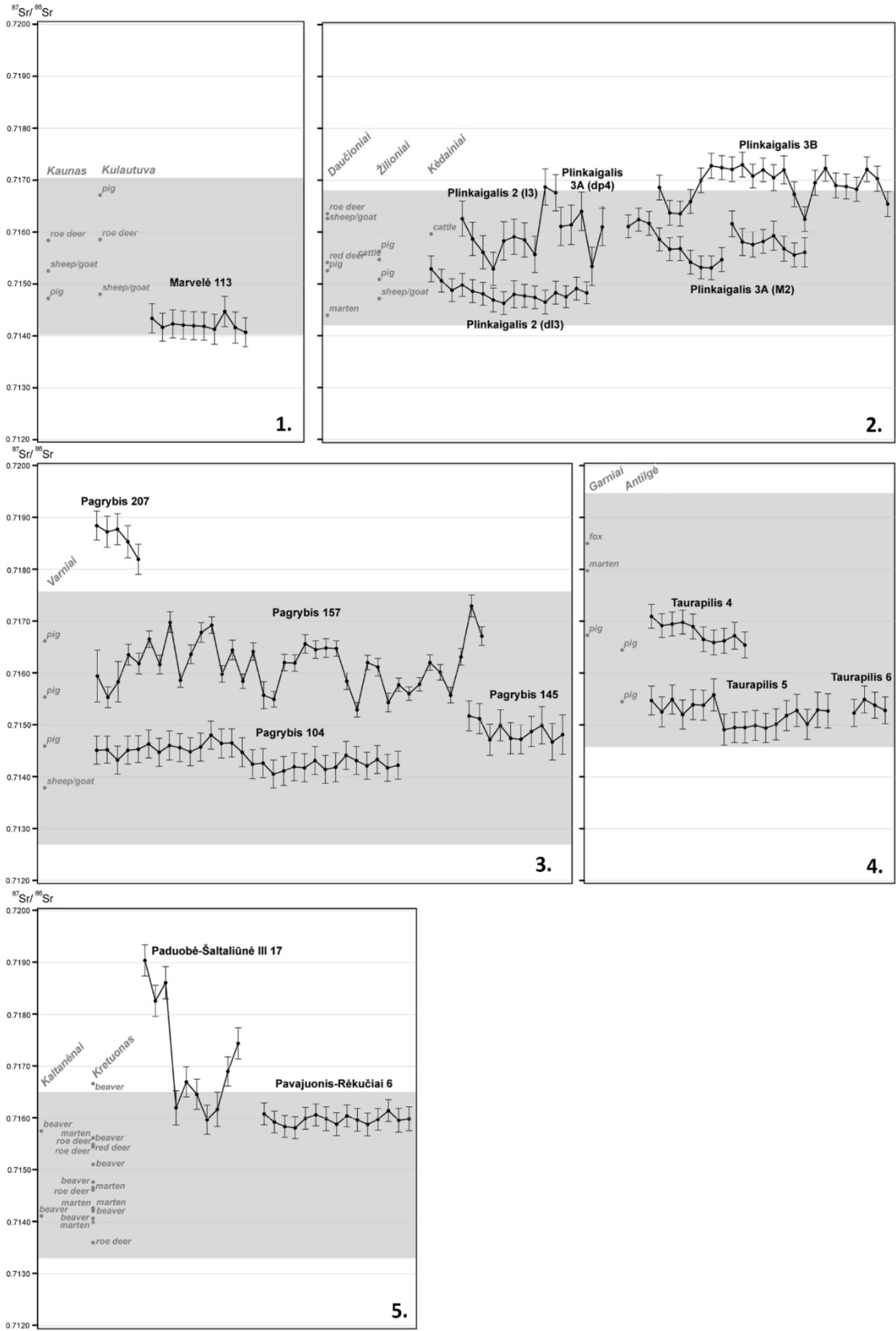

**Figure 2.** $^{87}Sr/^{86}Sr$ intra-tooth variation of the horses analyzed, plotted onto the local baselines, which were constructed from data obtained from archaeological animal specimens (shaded in grey; mean value ± 2 SD).

The $^{87}$Sr/$^{86}$Sr baseline, constructed from four animals from Varniai, for the Pagrybis cemetery is 0.7151 ± 0.0012 (2 SD) (Table 1 and Table S1: 18 and [20]). The horse buried in grave 104 was 132.3 cm tall, and its $^{87}$Sr/$^{86}$Sr values varied from 0.7141 to 0.7148, falling within the limits of the local baseline. The individual had a low mobility as indicated by a low $^{87}$Sr/$^{86}$Sr intra-tooth variation (SD = 0.0002). A smaller horse, 129.3 cm tall, was buried in grave 145 with a $^{87}$Sr/$^{86}$Sr range of 0.7147–0.7152, which is also within the local baseline. Although a cyclical pattern can be discerned in the individual's intra-tooth variation curve (Figure 2: 3), this individual was essentially quite sedentary (intra-tooth $^{87}$Sr/$^{86}$Sr SD = 0.0003), which was similar to horse 104. The $^{87}$Sr/$^{86}$Sr ratio (0.7153–0.7173) of the 127.3 cm tall horse from grave 157 was within the local baseline, as well. However, in contrast to horses 104 and 145, its intra-tooth variation was one of the highest among the horses analyzed (SD = 0.0005) and shows the comparatively high mobility of the horse. In addition, a cyclical pattern is visible in its intra-tooth variation curve (Figure 2: 3). The horse buried in grave 207 was the smallest of the four buried horses in Pagrybis, only 125.4 cm tall, and its five $^{87}$Sr/$^{86}$Sr values 0.7188–0.7182 are all above the local baseline (Figure 2: 3). They are higher than the values of any of the other horses and most of the analyzed humans buried in the cemetery [43]. The intra-tooth variation of the horse is SD = 0.0003, but only five measurements were carried out from its tooth. The horse also stands out from the other cemetery horses due to its enriched $\delta^{15}$N value (7.1‰) (Figure 4).

In order to develop the local $^{87}$Sr/$^{86}$Sr baseline (0.7155 ± 0.0013) for the Plinkaigalis cemetery, we analyzed 10 animal samples from three sites (Tables 1 and S1: 18) [20]. The horse found in grave 2 was 127 cm tall, and two of its teeth were analyzed (d$I_3$ and $I_3$). Although the $^{87}$Sr/$^{86}$Sr values of both teeth fall within the local baseline (Figure 2: 2), the $^{87}$Sr/$^{86}$Sr values of the deciduous tooth are lower than the permanent tooth (0.7146–0.7153 and 0.7153–0.7169, respectively). In addition, there is a significant difference between the intra-tooth variation of both teeth. The SD of the deciduous tooth is low (0.0002), whereas the permanent tooth is one of the highest (0.0005) among the horses analyzed (Figure 3). This difference leads to the conclusion that the pregnant mare and young horse later differed in mobility. Despite the different lifestyles, both individuals lived within the limits of the local baseline or in other localities with similar $^{87}$Sr/$^{86}$Sr signatures. Plinkaigalis horses 3A and 3B were buried inside a single grave. Horse 3A is only 1.5–2 years old, but was already 129.3 cm tall. Two of its teeth—d$P_4$ and $M_2$—were analyzed, i.e., data for this individual are available from the second half of the mare's pregnancy until its early death. The $M_2$ samples were taken from two different sides of the tooth. The mean $^{87}$Sr/$^{86}$Sr values were similar for the deciduous and the permanent tooth, 0.7160 ± 0.0002 and 0.7158 ± 0.0003, respectively (Table 2, Table S1: 12 and 13). Horse 3B was 137.3 cm tall and its intra-tooth curve coincided with the upper boundary of the local strontium range (Table 2, Figure 2: 2). Here, we concluded that this horse was non-local, due to the fact that 19 out of 23 measurements, including the first one, are outside of the local baseline and the $^{87}$Sr/$^{86}$Sr values are higher than in other horses as well as most of the analyzed humans buried at the site [43]. The intra-tooth variation of the horse displays a cyclical pattern, and its SD (0.0003) should be regarded as implying an average mobility (Figure 3).

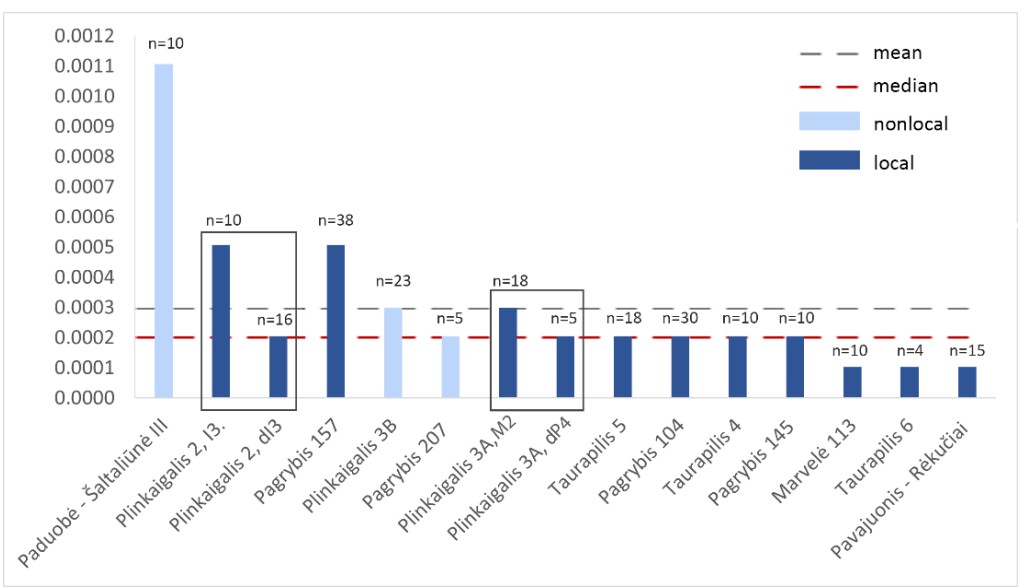

**Figure 3.** Intra-tooth variation and SD of the horses studied. Mean and median are calculated on all of the horse SD. The teeth of the same individual are defined. The number of measurements is indicated above the columns.

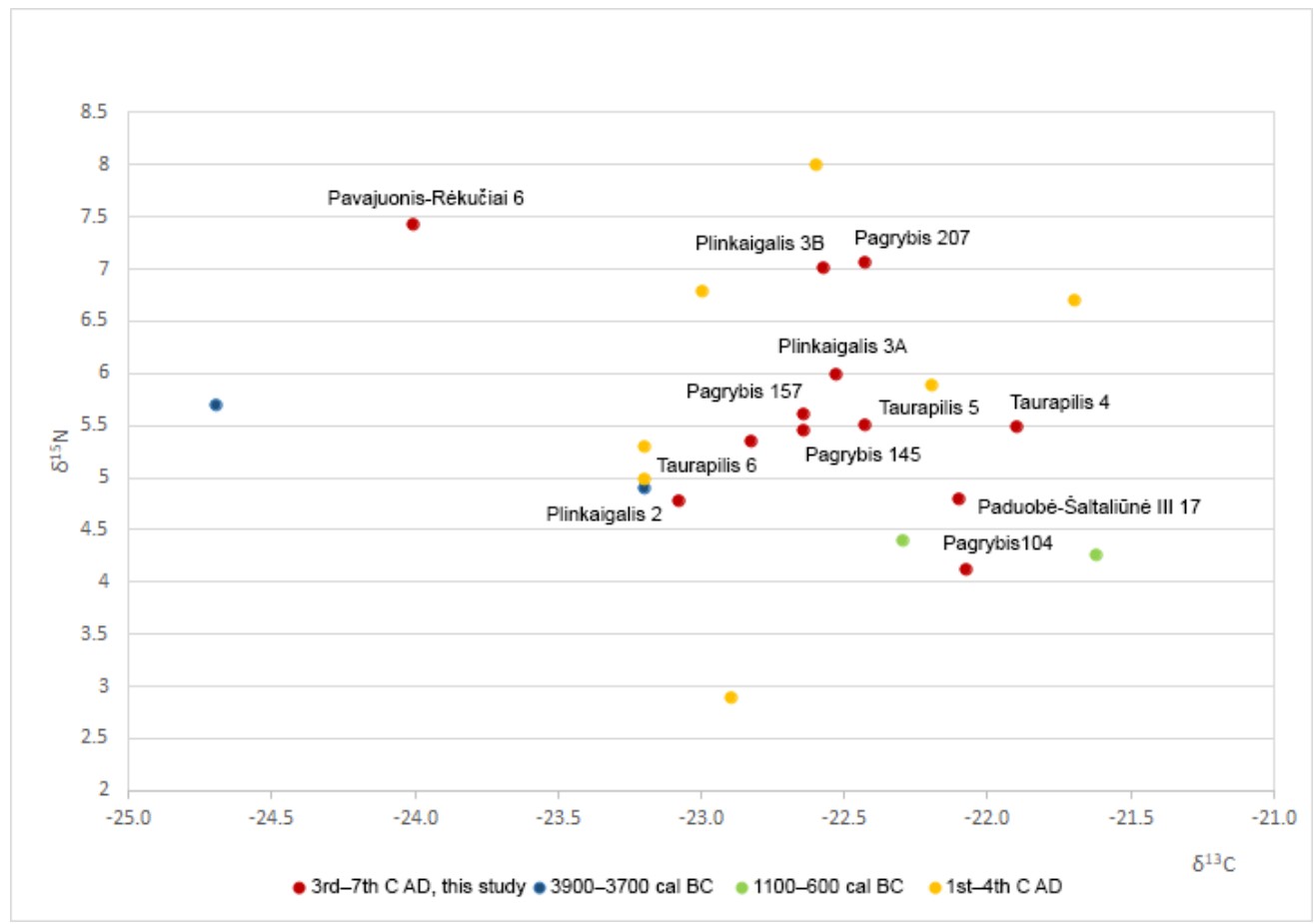

**Figure 4.** Carbon ($\delta^{13}$C) and nitrogen ($\delta^{15}$N) isotope values of measured bone collagen from Lithuanian horses. The horses measured in this study are dated to the 3rd–7th C AD (red symbols). Previously published horses by [44] (blue 3900–3700 cal BC and one green symbol 1100–600 cal BC) and [45] (one green symbol 1100–600 cal BC and the yellow symbols are dated to the 1st–4th C AD).

The local baseline for the Taurapilis barrow cemetery (0.7170 ± 0.0025, 2 SD) was constructed from measurements from animal samples (*n* = 5) from two different sites (Table 1). Then, the $^{87}$Sr/$^{86}$Sr ratios were determined for three horses. Horse 4 was 130.8 cm tall and its $^{87}$Sr/$^{86}$Sr values of 0.7165–0.7170 all fall within the local baseline. In barrows 5 and 6, two horses with similar osteometric data and age as well as similar stable isotope values were buried. They were 125–128 cm tall and 3–3.5 years old. The horse's strontium values are very similar (0.7149–0.7156) and are all within the local baseline (Figure 2), while their intra-tooth variation is smaller than average (SD 0.0001–0.0002). Their $\delta^{15}$N values (5.4‰ and 5.5‰) and the identical AMS $^{14}$C dates of the horses (Table 2) do not contradict the possibility that the horses are grazed in the same environment together. At the same time, the $^{87}$Sr/$^{86}$Sr ratios of horses 5 and 6 are lower, about 0.0015, than horse 4 (Figure 3). Therefore, horses 5 and 6 were definitely herded in a different area than horse 4.

## 4. Discussion

### 4.1. $^{87}$Sr/$^{86}$Sr Ratios

The results demonstrate that the $^{87}$Sr/$^{86}$Sr values of most of the analyzed horses (10 individuals, 76.9%) are within the local baselines (Table 2, Figure 2). However, three horses (23.1%) (Paduobė-Šaltaliūnė III, Pagrybis 207, and Plinkaigalis 3B) are of non-local origin. Two of these horses are the largest horses from the LRP–PMP in Lithuania with a withers height of 137–138 cm. Non-local ratios were indeed expected for the largest horses, as horses larger than local are usually associated with a non-local origin [12,13,46,47]. This hypothesis was also supported by the common migration processes that took place in Europe during this period, as well as by the repercussions of these processes seen in the archaeological records of Lithuania. Strontium isotope ratios (from ~0.716 to ~0.719) of all non-local horses were higher than the local baselines. Archaeological animals with these high $^{87}$Sr/$^{86}$Sr ratios are known from several other localities in Lithuania (e.g., Kupiškis, Garniai, and Pašatrijys [20]). However, similar ratios are also common in some more distant regions, e.g., Southern Sweden (Figure 5). Therefore, with the current data available, it is not yet possible to determine where the non-local horses originated from. The interpretation of the results is also hindered by the scarcity of data for the bioavailable $^{87}$Sr/$^{86}$Sr in Latvia, the Sambian Peninsula, and Northeast Poland, while in Belarus, the measurements of $^{87}$Sr/$^{86}$Sr have not been conducted at all and the local baselines can only be surmised based on geological information.

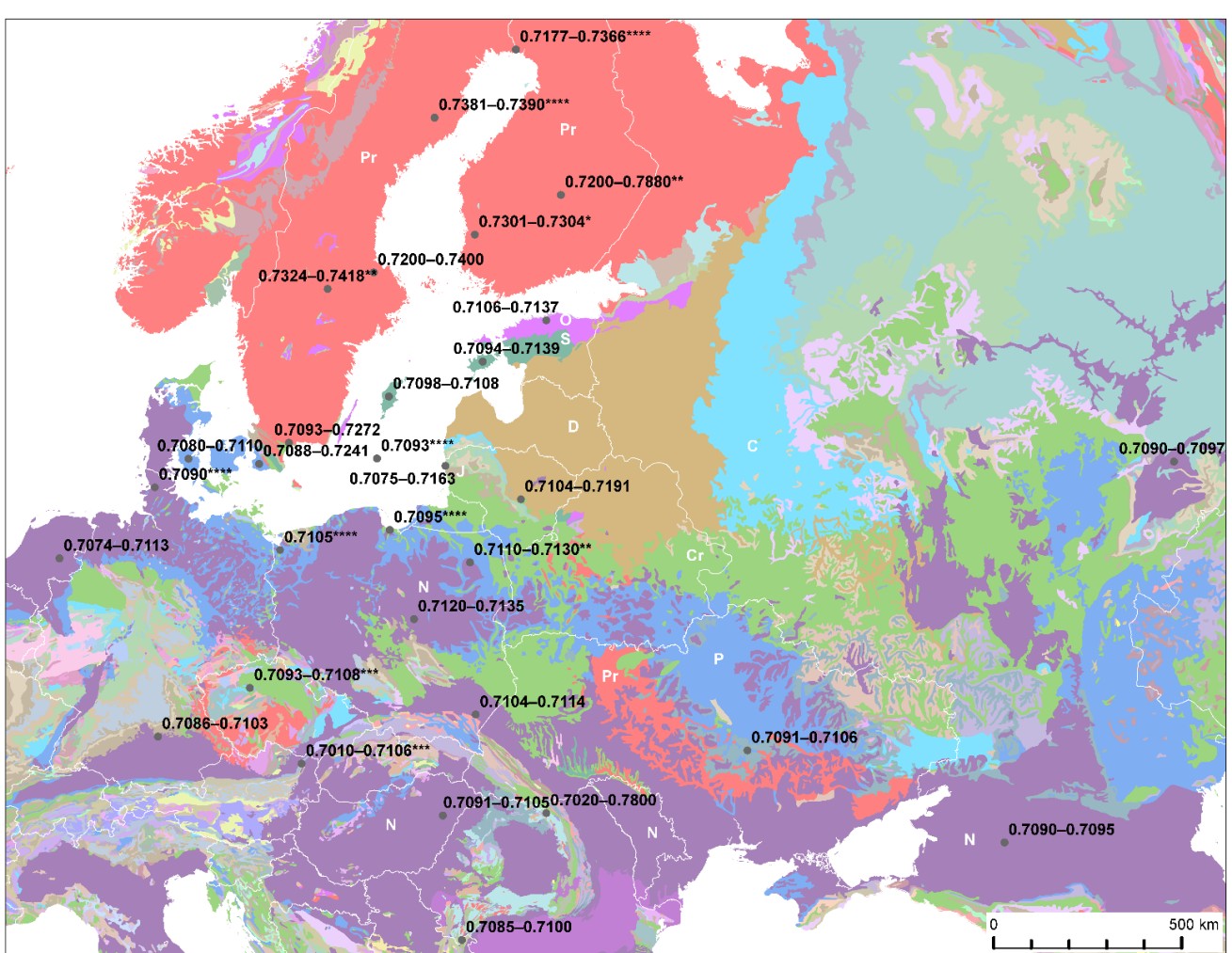

**Figure 5.** $^{87}$Sr/$^{86}$Sr ratios plotted on a geological map of Eastern and Central Europe. Geological layers have been taken from IGME5000 (2005). N—Neogene; P—Paleogene; Cr—Cretaceous; J—Jurassic; T—Triassic; C—Carboniferous; D—Devonian; S—Silurian; O—Ordovician; Pr—Precambrian. $^{87}$Sr/$^{86}$Sr ratios without a symbol are measured from archaeological fauna; * denotes ratios from archaeological human enamel; ** from ground water; *** from archaeological human dentine; **** from surface water [17,19,20,48–64].

The analysis of the grave goods does not permit an unequivocal definition of possible migration vectors. The wealthiest burial feature sets of grave goods originate from various regions and reflect long distance military campaigns or intensive multi-directional connections. Furthermore, radiocarbon dating necessitates a slight revision to the initially assumed chronology of some of the burials by assigning it possibly to the LRP or PMP, rather than the previous assignment to the MP (Hunnic Wars, fall of the Western Roman Empire, and the decades after the fall of the Hunnic Empire) [65]. With regards to the currently available data, at least two phases of horse burial tradition, dated to around 230–530 and 550–760 cal AD, can be distinguished. Some grave goods from earlier period burials confirm possible contacts with South Scandinavia [5,38,39]. However, this evidence is yet insufficient to define the origin of the riders as definitively originating from Southern Scandinavia. The largest proportion of the assemblage of grave goods would suggest links with the Germanic South or Central Europe [35]. It should be reckoned that during the periods of dynamic migrations, the route of migrants could have extended across different regions from Southern to Northern Europe, including those for which we do not yet have any $^{87}$Sr/$^{86}$Sr data. A greater mobility for these periods can also be inferred within the borders of Lithuania.

The latter phase of the horse burial tradition is reflected by burials from Pagrybis and Plinkaigalis. The source of this tradition could be associated with the Avars who buried both whole and dismembered horses [66–69]. The Olsztyn and Elbląg cultural groups acted as intermediaries in the spread of this practice [70]. Sacrifices of body parts of horses in bogs or in burials are also known from Southern Scandinavia until the 6th C AD, where they are linked to the Hunnic tradition [7,71–73]. Inhumations of whole horses, albeit rare, also occur during the Vendel Period (around 550–790 AD) [74–76]. Horses 130–140 cm tall, which in the Lithuanian context would be considered "large", are found in MP and VP burials in Southern Sweden [77]. Therefore, zooarchaeological data would not contradict the hypothesis that large horses were brought to Lithuania from Southern Sweden. Nevertheless, warrior graves of this period in Lithuania do not indicate an intense contact with Southern Scandinavia. Without excluding a possibility that some horses reached Lithuania from that region, we should not oversimplify the connection between the individual horses to the horse burial tradition. A horse of a South Scandinavian origin buried according to a nomadic tradition should not be considered as a big surprise. In order to find the origins of the latter, it is still necessary to identify the earliest horse graves in the region that would be contemporary to the emergence of the new burial custom.

### 4.2. Horse Herding

Another topic of discussion enabled by the strontium analyses is the practice of horse herding. With regards to the $^{87}Sr/^{86}Sr$ intra-tooth variation, the mobility of young horses and pregnant mares varied and could have been different between the individuals buried in the same cemetery. For instance, the Plinkaigalis 2 horse (a mare) lived in a different environment during its pregnancy and was less mobile than its descendant. Meanwhile, the $^{87}Sr/^{86}Sr$ of the latter are in complete agreement with horse 3A from the same cemetery, both during the time of birth and during the month 7–37 of its life (Table 2; Figure 2: 5). In general, the $^{87}Sr/^{86}Sr$ intra-tooth variation was rather diverse. For 11 (73.3%) individuals, it was lower than the average. In addition, for 8 (53.3%) of them, it was lower than the median (Figure 3). These results indicate that most of the horses have been herded in the area surrounding the cemeteries, i.e., they were bred in the local communities. Cyclical patterns can be discerned in the intra-tooth variation of some of the horses (e.g., Pagrybis 157, 145; Pavajuonis-Rėkučiai). Others display slight but permanent $^{87}Sr/^{86}Sr$ changes (e.g., Pagrybis 104, Taurapilis 5), which can be linked to the changing of pastures. The cases mentioned are seen in the intra-tooth variation of young horses, which are not older than ~4 years. Horses of this age were only beginning to be ridden [12,78]. Therefore, on the one hand, the periodicity or small, long-term changes in $^{87}Sr/^{86}Sr$ should not be associated with the rider's journeys. On the other hand, horses, including young individuals, may have been driven and their mobility may reflect the rider's trip.

In some instances, a clear drop can be seen in the horse intra-tooth $^{87}Sr/^{86}Sr$ values (e.g., Paduobė-Šaltaliūnė III). Although a drop is also apparent in Pagrybis 207, only five measurements were made for this horse, and this does not allow for a more complete picture (Figure 2). Both horses are non-local and the marked drop $^{87}Sr/^{86}Sr$ values demonstrate that they relocated at the time when mineralization of the tooth enamel was still taking place. Tooth $P_2$ was analyzed for both horses, and its enamel is mineralized during the months 13–31. Therefore, judging by the intra-tooth variation curves, both horses arrived in the environment of the cemeteries in the second year of their lives. For free living horses, weaning takes place naturally at around 8–9 months or a small amount of time later [79,80]. Accordingly, we may presume that the young horses were moved to new pastures not long after they were fully weaned.

Although the horses from the same burial site had local $^{87}Sr/^{86}Sr$ ratios, their values were rarely identical (e.g., Taurapilis 4 and 5), which indicates that the horses lived in different environments. Due to the differences in their function and status, the horses were most probably not maintained and fed in the same way. For example, some could have been left free-roaming in the forests, while others were herded close to the houses.

Moreover, the horses could have been herded by several separate communities which used the same burial site or they could have participated in local migrations that are still within the baseline. There is little information on how the MP or VP horses were herded. As the cases from the VP and later periods in Iceland and Scandinavia demonstrate, some of the horses were free-roaming. The frequently used work horses, in contrast, remained close to home [81]. Certainly, both the food rations and mobility varied among horses of different functions. The scarcity of pastures and the fact that free-roaming horses almost did not require human care were the reasons for allowing them to free-roam [81]. In the 1st millennium AD, the territory of Lithuania was probably characterized by a shortage of pastures, which lasted until as late as the 20th C AD [82]. It is very likely that in the 1st millennium AD horses were left free-roaming in the territory of current Lithuania, as well. There are records regarding free-grazing horses in the Early Modern Period in Lithuania in written sources dating to the 16th C AD. With regards to the Second Statute of Lithuania (1566), herds of horses were allowed to graze without a herdsman until Saint George's day (23 April) and again from 1 October. During the period in between, the horses were to be attended by a herdsman [83]. In the early 20th C AD, horses grazed freely in the Curonian Spit (coastal Lithuania, previously the Kingdom of Prussia). They would wander for 25–30 km away from the villages [84]. Horses were also free-roaming on the island of Gotland and in Norway as late as in the 19th C AD [81] and are still in Iceland. The horse's lifestyle and diet were determined not only by their function, but also by their sex. Medieval and Early Modern Period sources from North and East Europe mention herds of mares and foals roaming in semi-wild conditions, often in wooded areas [83,85–87]. In the 13th–14th C AD, herds of mares are mentioned in Prussia, lands occupied by the Teutonic Order, and also in Lithuania. Both Lithuanians and Prussians used to steal herds of mares and other horses from the Order and vice versa [10,78,88]. Therefore, it is likely that free-roaming herds of mares and foals existed during the LRP–PMP, as well. The diet and mobility of the horses from these herds should have been different from those of the horses which were ridden, and thus remained close to home. Perhaps these differences have resulted in the twice as low intra-tooth variation (SD = 0.00016) of pregnant mares compared to the young horses (0.00032).

### 4.3. Horse Diet

The $\delta^{13}C$ and $\delta^{15}N$ values of the analyzed horses suggest a relatively homogenous diet based on $C_3$ plants. However, in some cases, $\delta^{13}C$ or $\delta^{15}N$ slightly deviate from the other horses (Figure 4; Table 2). One of the most notable outliers is the horse from Pavajuonis-Rėkučiai. This 5–7 year-old horse has a significantly more depleted $\delta^{13}C$ (−24.0‰) and more enriched $\delta^{15}N$ (7.4‰) compared to the average $\delta^{13}C$ (−22.7‰ ± 0.5) and $\delta^{15}N$ (5.7‰ ± 1.0) of all the analyzed horses. It is very likely that this animal was pastured in a forest with a thicker canopy in which plants, due to the so-called canopy effect, tend to have more depleted $\delta^{13}C$ values [89]. The enriched $\delta^{15}N$ value of this horse could have resulted from a comparatively lower moisture content in the soil where the $C_3$ plants that the animal was feeding on grew [90]. Therefore, the horse from Pavajuonis-Rėkučiai may have been pastured in dry pine forests which largely grew on the glaciofluvial sands of the Žeimena River upper basin, where the SD of $^{87}Sr/^{86}Sr$ for Pavajuonis-Rėkučiai horse was the lowest (0.0001) compared to all of the analyzed horses. Likely, this horse had a low mobility during the second and third years of its life. The Pavajuonis-Rėkučiai horse together with two other horses (Pagrybis 207 and Plinkaigalis 3B) had exceptionally enriched nitrogen values, above 7‰. The latter two horses are non-local, whereas the Plinkaigalis 3B horse is one of the largest horses from this period (137 cm). The plant $\delta^{15}N$ increases through fertilization [91,92]. Therefore the enriched $\delta^{15}N$ in horses could reflect additional fodder and probably a better diet [90,93]. Alternatively, these horses could have been herded close to the settlement in pastures fertilized by the manure of pastured animals.

## 5. Conclusions

The $^{87}$Sr/$^{86}$Sr analysis of Lithuanian horses partially supports the hypothesis that larger horses dating to the LRP–PMP are of non-local origin. We identified 3/13 (23.1%) non-local horses, of which 2/3 (66.7%) were the largest individuals from the MP and PMP in Lithuania. However, we cannot reject the fact that other horses larger than the typical local ones could be the second or later generation descendants of non-local horses or that they originated from foreign regions with $^{87}$Sr/$^{86}$Sr similar to Lithuania. Furthermore, as one case demonstrates, non-local horses occasionally do not stand out in size. Unfortunately, the overlapping ranges of bioavailable $^{87}$Sr/$^{86}$Sr in most of the regions of Europe limit our ability to establish whether non-local horses have originated from other areas in Lithuania or from more distant regions. With regards to the $^{87}$Sr/$^{86}$Sr data, the area of origin of the non-local horses could be Southern Sweden. Osteometric data of the horses do not contradict this hypothesis, while archaeological data can only partially support it. This provides a basis for a new discussion on trends in migration during this transitional period. It also compels us to rethink the current models which situate Southern and Central Europe as the dominant vectors of migration. The direct identification of the origin of individual horses based on the influences observed in burial traditions is very simplistic, as the latter may be a continuation of earlier processes. At this stage, we cannot speculate on the number of non-native horses and their impact on native local horses. Possibly, these questions will be successfully answered by future DNA research. The analyses of $^{87}$Sr/$^{86}$Sr, $\delta^{13}$C, and $\delta^{15}$N demonstrate that horses buried in a cemetery of the same community had different mobility and feeding patterns. The observed differences could also be related to the different function and sex of the horses as well as the lifestyle of their owners. The least mobile horses were pregnant mares, while the extremely enriched $\delta^{15}$N of 3/13 horses (2/3 non-local) may be associated with an exceptional and, likely better, diet.

**Supplementary Materials:** The following supporting information can be downloaded at https://www.mdpi.com/article/10.3390/heritage5010018/s1. Table S1: $^{87}$Sr/$^{86}$Sr measurements on teeth enamel of horses and other animals.

**Author Contributions:** Conceptualization, G.P. (Giedrė Piličiauskienė) and L.K.; data curation, E.S. and M.K.-S.; investigation, G.P. (Gytis Piličiauskas), L.K., E.S., and G.P. (Gytis Piličiauskas); methodology, K.L., E.K., and M.K.-S.; visualization, G.P. (Giedrė Piličiauskienė), L.K., and G.P. (Gytis Piličiauskas); writing—original draft, G.P. (Giedrė Piličiauskienė) and L.K.; writing—review and editing, K.L. and E.K. All authors have read and agreed to the published version of the manuscript.

**Funding:** This research was financially supported by the Research Council of Lithuania (S-MIP-19-67) and partly supported by Vegacenter.

**Institutional Review Board Statement:** Not applicable.

**Informed Consent Statement:** Not applicable.

**Acknowledgments:** Translation of this article was funded by the Juozas Sidas Foundation. We would also like to thank the editors of this special volume for the invitation to contribute and two anonymous reviewers whose comments improved the final version of the article. This is Vegacenter publication number #47.

**Conflicts of Interest:** The authors declare no conflict of interest.

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
