# Peer review of "The Origin of Late Roman Period–Post-Migration Period Lithuanian Horses"

_heritage, doi:10.3390/heritage5010018_

Round 1

Reviewer 1 Report

This is a very well prepared article, with a very important issue. Results are reliable, and data and their interpretation give a good base for further studies on horse provenience in the Baltic region.

Please, consider a few comments given below for improvement in some details in the article. 

In the introduction the statement, that horses in Lithuania were unusual large with the average size of 129,8 cm is exaggerated. With the comparison to other regions in Europe, they were smaller (see Benecke N. 1994. Archaozoologische Studien zur Etwiklung der Haustierhaltung in Mitteleuropa und Sudskandinavien von den Anfanggen bis zum ausgesagen Mittelalter).

The weak side of the article is no 14C dates. This is obvious, that dates of skeleton deposits do not always correspond to chronology on the basis of archaeological context and/or archaeological findings (artefacts). It is advised to give some explanation why you missed the radiocarbon dates. Please, note that the horse skeleton from Marvele is not related to human burials, and the time span in burial is really long because of 2nd - 12th CAD. How did you find out, that the horse was buried in any period from the time span? It could be from any other one. 

In table 2 would be positive to add the column with indication to which period are belonging skeletons.

Please, correct the column of the withers height in table 2. Values given in it are in several horses different than provided in the text.

There was no chance to check the reference, because in the file for review it was missed.

Author Response

  1. In the introduction the statement, that horses in Lithuania were unusual large with the average size of 129,8 cm is exaggerated. With the comparison to other regions in Europe, they were smaller (see Benecke N. 1994. Archaozoologische Studien zur Etwiklung der Haustierhaltung in Mitteleuropa und Sudskandinavien von den Anfanggen bis zum ausgesagen Mittelalter).

Response 1: Clarified and corrected

  1. The weak side of the article is no 14C dates. This is obvious, that dates of skeleton deposits do not always correspond to chronology on the basis of archaeological context and/or archaeological findings (artefacts). It is advised to give some explanation why you missed the radiocarbon dates. Please, note that the horse skeleton from Marvele is not related to human burials, and the time span in burial is really long because of 2nd - 12th CAD. How did you find out, that the horse was buried in any period from the time span? It could be from any other one. 

Response 2: All analysed horses were radiocarbon dated and AMS 14C dates were presented in Table 2. We apologies, that this remained unclear. We have added an additional explanation in the text regarding the direct dating and referred Table 2

  1. In table 2 would be positive to add the column with indication to which period are belonging skeletons.

Response 3. AMS 14C dates of every horse are presented in Table 2 in separate column, chronology of all periods was described in the text, therefore, we don’t think column with indication about the period is necessary (see response for point 2 above, please)

  1. Please, correct the column of the withers height in table 2. Values given in it are in several horses different than provided in the text.

Response 4. Corrected

  1. There was no chance to check the reference, because in the file for review it was missed.

Response 5. Apologies... misunderstanding, references were added to another file….

Reviewer 2 Report

I want to congratulate the authors.  The article is an excellent one in nearly all ways -- with the exception that no references were provided.  The text is relatively clean; the research methodology is clearly presented; the datasets are detailed and admirably transparent; and the tables and figures appropriately captioned.  Corrections are minimal, but they do need to be made.  They are as follows:

Punctuation needs to be corrected in places.

Millennium needs to be millennia.

Subject/verb agreement needs to be corrected in places.

Incorrect grammar needs to be remedied in places.

Despite the need for very few corrections, this is a masterful article and provides important information that is significant for the field.

Author Response

  1. Punctuation needs to be corrected in places.

Response 1. Corrected

  1. Millennium needs to be millennia.

Response 2. Corrected

  1. Subject/verb agreement needs to be corrected in places.

Response 3. Corrected

  1. Incorrect grammar needs to be remedied in places.

Response 4. Corrected